# Mutagenesis of Odorant Receptor Coreceptor *Orco* Reveals the Odorant-Detected Behavior of the Predator *Eupeodes corollae*

**DOI:** 10.3390/ijms242417284

**Published:** 2023-12-09

**Authors:** Ji-Nan Wu, Chen-Xi Cai, Wen-Biao Liu, Dong Ai, Song Cao, Bing Wang, Gui-Rong Wang

**Affiliations:** 1State Key Laboratory for Biology of Plant Diseases and Insect Pests, Institute of Plant Protection, Chinese Academy of Agricultural Sciences, Beijing 100193, China; wujinan107@163.com (J.-N.W.); 16601177063@163.com (C.-X.C.); 82101221213@caas.cn (W.-B.L.); as26@foxmail.com (D.A.); csong@ccnu.edu.cn (S.C.); 2Shenzhen Branch, Guangdong Laboratory for Lingnan Modern Agriculture, Genome Analysis Laboratory of the Ministry of Agriculture, Agricultural Genomics Institute at Shenzhen, Chinese Academy of Agricultural Sciences, Shenzhen 518120, China

**Keywords:** *Eupeodes corollae*, Orco, in situ hybridization, gene editing, EAG, olfactory behavior

## Abstract

The successful mating of the hoverfly and the search for prey aphids are of great significance for biological control and are usually mediated by chemical cues. The odorant receptor co-receptor (*Orco*) genes play a crucial role in the process of insect odor perception. However, the function of *Orco* in the mating and prey-seeking behaviors of the hoverfly remains relatively unexplored. In this study, we characterized the *Orco* gene from the hoverfly, *Eupeodes corollae*, a natural enemy insect. We used the CRISPR/Cas9 technique to knock out the *Orco* gene of *E. corollae*, and the *EcorOrco^−/−^* homozygous mutant was verified by the genotype analysis. Fluorescence in situ hybridization showed that the antennal ORN of *EcorOrco^−/−^* mutant lack Orco staining. Electroantennogram (EAG) results showed that the adult mutant almost lost the electrophysiological response to 15 odorants from three types. The two-way choice assay and the glass Y-tube olfactometer indicated that both the larvae and adults of hoverflies lost their behavioral preference to the aphid alarm pheromone (*E*)-β-farnesene (EBF). In addition, the mating assay results showed a significant decrease in the mating rate of males following the knock out of the *EcorOrco* gene. Although the mating of females was not affected, the amount of eggs being laid and the hatching rate of the eggs were significantly reduced. These results indicated that the *EcorOrco* gene was not only involved in the detection of semiochemicals in hoverflies but also plays a pivotal role in the development of eggs. In conclusion, our results expand the comprehension of the chemoreceptive mechanisms in the hoverflies and offers valuable insights for the advancement of more sophisticated pest management strategies.

## 1. Introduction

Insects heavily depend on their exquisitely sensitive olfactory system to carry out a range of crucial activities. These activities encompass locating host plants and prey, identifying mates, and selecting suitable habitats and oviposition sites, as well as evading natural enemies [1,2,3]. The antennae of insects serve as the primary sensory organs for detecting olfactory signals from the external world, and they feature a multitude of diverse sensilla types [4]. The function of sensilla is determined by the specific olfactory receptor neurons (ORNs). The process of odorant recognition in insects is intricate. Initially, odorant molecules gain entry into the sensilla through epidermal pores and subsequently bind to odorant-binding proteins (OBPs). These molecules then travel through the lymphatic fluid of sensilla to reach the dendritic membrane of ORNs. There, they activate odorant receptors (ORs) expressed in ORNs, converting external chemical signals into electrical signals transmitted to the central nervous system [5,6]. Simultaneously, odorant-degrading enzymes (ODEs) rapidly degrade chemical signals to inactivate them, preventing constant neuronal stimulation and restoring the sensitivity of neurons to odor recognition [7].

In the peripheral olfactory recognition system, the ORs of the insect play a pivotal role in the detection of volatile odorants and the initiation of signal transduction. The insect ORs can be divided into two categories: one is the odorant receptor co-receptor (Orco), which generally has only one for each insect and exhibits a high degree of conservation among various insect species [8]; the other is the traditional odorant receptor (ORx), which varies significantly in number among different insect species and exhibits lower levels of homology [9,10]. The insect ORs form a heteromeric complex with the Orco, which functions as a non-selective cation channel [11,12].

In 2018, a groundbreaking study of the Orco protein in *Apocrypta bakeri* was conducted using cryo-electron microscopy. This study revealed that the Orco protein forms a homotetrameric channel structure, with the four subunits symmetrically arranged around the central pore. Interestingly, the regions of various ORs where the sequence is less conserved are predominantly situated in the central pore and anchor domain of this tetrameric structure [13]. In the OR/Orco heteropolymer, ORs take on the responsibility of binding to odorants and determining the ligand specificity, while Orco plays a pivotal supporting role in this process [14]. Functional studies have revealed that, in the absence of Orco, the ORs cannot be assembled and functioned, resulting in the loss of olfactory recognition mediated by ORs. Thus, the *Orco* gene has a central role in this process [15]. For example, when the *Orco* gene of *Drosophila melanogaster* was knocked out, the sensitivity of insects to the detection of odorant molecules was affected. However, the detection of odorant molecules by insects returned to normal after *Orco* gene rescue, which further confirmed the importance of *Orco* gene in olfactory recognition [16]. In *Ooceraea biroi*, the CRISPR/Cas9-mediated knockout of the *Orco* gene resulted in the loss of numerous glomeruli in the brain and induced changes in social behavior [17]. Similarly, after *Orco* gene knockout, *Heliothis armigera* has olfactory defects, which affects the chemical information exchange and oviposition behavior. The male *Spodoptera frugiperda* lost the ability to recognize the sex pheromone released by the female moth and could not mate [18].

Hoverflies can use different chemical cues such as aphid alarm pheromones, herbivore-induced plant volatiles (HIPVs), and aromatic volatiles to locate prey and evaluate food suitability. Some chemosensory mechanisms have been reported. Li et al. (2020) found that *EcorOR25* of *Eupeodes corollae* was involved in the recognition of aromatic volatiles such as *p*-cresol, eugenol, and methyl eugenol, thereby mediating attractive behavior [19]. In a separate study, Wang et al. (2022) delved into the chemosensory mechanisms by which *E. corollae* senses the (*E*)-β-farnesene (EBF) of the aphid alarm pheromone component and accurately locates the aphid, demonstrating that *EcorOR3* plays a key role in the detection of EBF by the aphid predator *E. corollae* [20]. Recent studies have revealed that *EcorOR4* can be activated by 1-octen-3-ol and significantly attract mated females [21]. These olfactory recognition processes are closely intertwined with the function of Orco. The role of the *Orco* gene in the life activities of *E. corollae*, especially in locating host plants and prey, finding spouses and selecting oviposition sites, needs further study. 

In this study, we used CRISPR/Cas9 technology to knock out the *EcorOrco* gene, subsequently confirming the *EcorOrco^−/−^* homozygous mutant through fluorescence in situ hybridization. Following this, a series of experiments were conducted to elucidate the pivotal role of the *Orco* gene in various aspects of *E. corollae* behavior: (1) the electroantennogram responses to 15 odorants from different sources were compared; (2) the behavioral responses of *E. corollae* to the aphid alarm pheromone EBF were compared using the glass Y-tube olfactometer and the two-way choice assay; (3) the mating rate was compared; (4) the egg production and egg hatching rates were compared. Our results demonstrate that the *EcorOrco* gene plays an important role in a variety of life activities of the *E. corollae*. It provides a functional basis for exploring the olfactory recognition of the hoverfly mediated by odorant receptors. Furthermore, it provides a theoretical basis for screening the stable and efficient regulators of natural enemy insect behavior.

## 2. Results

### 2.1. Generation of EcorOrco^−/−^ Mutant Line by CRISPR-Cas9 Technique

To elucidate the role of the *Orco* gene in *E. corollae* olfaction, we utilized CRISPR-Cas9 technology to knock out the *EcorOrco* gene. We designed two sgRNAs, targeting exon 2 and exon 3, in order to achieve a long fragment deletion in the DNA sequence (Figure 1A). The two specific sgRNAs and Cas9 protein mixture were co-injected into 1300 freshly laid eggs. Of these, nearly 53 eggs successfully hatched and 29 larvae survived until adulthood. These individuals were allowed to intermingle. The mated adults were reared in a single tank and provided with broad bean plants with aphids for oviposition. After completing the oviposition, the genomes of the G0 generation’s females and males were examined. Offspring of the same genotype were raised together. Following five generations of selective screening, we successfully obtained a line with a 637 bp DNA fragment deletion in the genome (Figure 1A,B). Notably, the first three transmembrane domains were deleted and 1 amino acid was deleted in the fourth transmembrane domain, resulting in a total deletion of 160 amino acids (Figure 1C).

### 2.2. Morphology of Antennal ORNs in EcorOrco^−/−^ Mutants

To test whether *EcorOrco*^−/−^ mutations cause a loss of the full-length Orco protein and change the distribution of ORNs, we performed immunostaining on the antennal sections of both wild-type (WT) and *EcorOrco^−/−^* mutant *E. corollae* using an *Orco* antisense probe and counterstained the cell nuclei with DAPI (Figure 1D). As predicted, the results of the cell nuclei staining showed that the Orco gene was densely expressed in the antennal ORNs of WT hoverflies and the antennae of the *EcorOrco^−/−^* hoverfly lack Orco staining, indicating that the full-length Orco protein is absent. However, the majority of ORNs in the antenna of *EcorOrco^−/−^* mutants are still present just like the antennal ORNs in WT adults, as indicated by DAPI signals, suggesting that the loss of the *EcorOrco* gene did not affect the distribution of ORNs in *E. corollae* (Figure 1D).

### 2.3. EcorOrco^−/−^ Mutants Lose the EAG Responses to Almost All Tested Chemical Compounds

To assess the impact of the *EcorOrco* gene on the peripheral responses across the antennae of *E. corollae*, we compared the electrophysiological responses on both the WT and mutant individuals using EAG tests. This experiment uses a broad odorant panel of 15 compounds ranging across three classes, namely aromatic, aliphatic, and terpene compounds (Table 1). The results indicated that the WT males and females showed robust EAG responses to the 15 volatiles (Figure 2A,B). Conversely, the *EcorOrco^−/−^* mutants displayed a nearly complete loss of EAG responses to the tested chemicals, except for 2,3-butanedione and 3-methyl-2-butenal (Figure 2C,D). Compared to the WT of both sexes, the *EcorOrco^−/−^* mutants did not significantly reduce the EAG response to 2,3-butanedione (Figure 2), suggesting that there are other olfactory pathways involved in its detection.

### 2.4. Behavioral Choice Assays for EcorOrco^−/−^ Mutants

Previous studies have demonstrated that EBF can attract both the adults and larvae of *E. corollae * [18]. In this study, in order to elucidate the pivotal role of the *Orco* gene in the detection of EBF by *E. corollae*, we compared the behavioral choices of WT and *EcorOrco^−/−^* mutant of the *E. corollae* in the adult and larval stages to EBF. The results indicated that the WT larvae exhibit a strong preference for EBF, whereas *EcorOrco^−/−^* mutants do not exhibit any preference (Figure 3A). The Y-tube olfactometer assays were then performed with mutated adult hoverflies of both sexes. In contrast to the strong attraction of WT individuals to EBF, mutant hoverflies of both sexes displayed no preference to EBF (Figure 3B). These results demonstrate that both larva and adult *EcorOrco^−/−^* mutants had completely lost their behavioral preference to EBF.

### 2.5. The Mating Behavior of the EcorOrco^−/−^ Mutants

To study the effect of the *Orco* gene on the adult mating behavior, we conducted a comparative analysis of mating rates involving four different mating combinations. The results indicated that there was no significant difference of the mating rate between female WT × male WT (FWT × MWT, 57.1%) group and female homozygotes × male WT (FHo × MWT, 50%) group (*p* = 0.549) (Figure 4). However, it is noteworthy that, out of 30 pairs in female WT × male homozygotes (FWT × MHo) group, only four pairs were able to successfully mate. The mating rate of homozygotes significantly decreased when compared to that in FWT × MWT group (*p* < 0.05) (Figure 4). Similarly, we observed that, out of 40 pairs in female homozygotes × male homozygotes (FHo × MHo) group, only four pairs were able to successfully mate, also showing a significant decrease in the mating rates compared to that in FWT × MWT group (*p* < 0.05) (Figure 4). This result indicate that the *Orco* gene plays an important role in the male mating process.

### 2.6. The Fecundity and Hatching Rate of the EcorOrco^−/−^ Mutants

To further study the role of the *Orco* gene on fecundity and hatchability, we compared the egg-laying numbers and the hatching rates in four different mating combinations. The results indicated that the egg-laying numbers of females in FWT × MWT, FHo × MWT, FWT × MHo, and FHo × MHo groups were 407.17, 55.58, 80.50, and 22.75 eggs in average (Figure 5A), and the hatching rates were 88.20%, 40.99%, 20.52%, and 14.19%, respectively (Figure 5B). There are significant differences in egg-laying numbers and the hatching rates between the FWT × MWT group and FWT × MHo group (*p* < 0.05), as well as between the FHo × MWT group and the FHo × MHo group (*p* < 0.05) (Figure 5A,B). These differences may be attributed to a decrease in sperm motility resulting from the knockout of the *EcorOrco* gene. Furthermore, significant differences were also observed in the egg-laying numbers between FWT × MWT group and FHo × MWT group (*p* < 0.05), as well as between FWT × MHo group and FHo × MHo group (*p* < 0.05) (Figure 5A). Additionally, the hatching rates of the FWT × MWT group and FHo × MWT group were significantly different (*p* < 0.05). The results showed that the deletion of the *EcorOrco* gene probably affected the normal development of embryos in females.

## 3. Discussion

Hoverflies have been widely used in pest management as biological control agents. Their olfactory system plays an important role in various aspects of their life activities, such as locating host plants and prey as well as finding mates and oviposition sites. However, the underlying mechanisms are largely unknown. In this study, we characterized the function of *Orco* in *E. corollae* and revealed the basic mechanism of its olfactory perception. The *EcorOrco* gene was knocked out using CRISPR/Cas9-mediated genome editing technology, which has been proven effective for genome editing in *E. corollae * [20]. After five generations of screening, we obtained the *EcorOrco^−/−^* mutant line with large fragment deletion. Although the first three transmembrane domains of *EcorOrco^−/−^* mutant are completely deleted, its nucleotide sequence can still be translated (Figure 1C). However, fluorescence in situ hybridization demonstrated the loss of *Orco* signals in the antennae of the *EcorOrco^−/−^* mutants. The EAG responses of the *EcorOrco^−/−^* mutant male and female hoverflies to the tested aphid alarm pheromone compound, HIPVs, and floral scents were severely weakened. These results suggest that the first three transmembrane domains of EcorOrco may be indispensable to form the correct structure of the heteromeric complex with ORx [18], and therefore the absence of the first three transmembrane domains might lead to alterations in the functionality of the OR–Orco complex, potentially influencing the cellular trafficking of the receptor to the plasma membrane [13]. 

Interestingly, *EcorOrco^−/−^* mutants of both male and female hoverflies still exhibited responses to several of the tested chemicals, such as 3-methyl-2-butenal and 2,3-butanedione (Figure 2C,D). This suggests that, in addition to the OR-dependent olfactory pathways, there may be other pathways in hoverflies, such as the ionotropic receptors (IRs), which play a role in recognizing these compounds [22]. This is consistent with studies in *D. melanogaster*, where single sensillum recording (SSR) recording showed that 2,3-butanedione not only activates ORs, but also elicits electrophysiological responses from IRs [23].

Natural enemies usually use chemical cues to accurately locate prey, such as kairomone HIPVs and pheromones released by prey [24]. In the trophic interactions, the most concerned compound EBF is not only an aphid alarm pheromone but also an HIPV. Previous studies have shown that *EcorOR3* is a key odorant receptor that mediates the detection of EBF by adults and larvae. However, the electrophysiological response of adults *EcorOR3^−/−^* mutant hoverflies to EBF was significantly decreased, but not completely lost, indicating that there may be other receptors involved in the recognition of EBF by *E. corollae* [18]. In this study, we found that the electrophysiological response of *EcorOrco^−/−^* mutant adults to EBF was fully lost (Figure 2C,D), and behavioral assays showed that the adults and larvae *EcorOrco^−/−^* mutants completely lost their behavioral preference to EBF (Figure 3A,B). This result demonstrates that the detection of EBF by *E. corollae* is mediated by OR-dependent olfactory pathways. Likewise, among the four parasitoid species (*Leptopilina heterotoma*, *Leptopilina boulardi*, *Leptopilina syphax*, and *Leptopilina drosophila*), female parasitoid wasps lacking the *Orco* gene exhibited diminished host-searching capabilities [25]. These results suggest that the *Orco* gene plays a key role in the localization of prey by their natural enemies.

Previous studies show the importance of OR-mediated olfactory recognition pathways in insects’ mating behavior. For example, the *DmelOR69a* of male *D. melanogaster* was involved in the detection of the sex pheromone (Z)-4-undecenal released by females, and can promote the long-distance communication before mating [26]. Moreover, *Helicoverpa armigera* employs the pheromone antagonist *cis*-11-hexadecenol to deter nonoptimal mating by inhibiting male individuals, in which *HarmOR16* played a crucial role [27]. In this study, we found that the mating behavior of *EcorOrco^−/−^* mutant males was significantly reduced, and only a few *EcorOrco^−/−^* mutant males were able to mate with females (Figure 4). However, the *EcorOrco^−/−^* mutated females were still able to mate with WT males (Figure 4). This suggests that the ability of males to detect sex pheromone cues is likely essential for locating females and producing mating behavior. Similar results have also been reported in Lepidoptera insects, such as *S. frugiperda* and *Heliothis virescens* [18,28].

The oviposition selection of gravid females is very important for offspring. Gravid females can detect various oviposition cues in the environment through visual, olfactory, and gustatory means, and finally make oviposition decisions to provide the most favorable living conditions for offspring [29,30,31,32]. Olfactory senses play a decisive role in this process. Insects can use a precise olfactory system to distinguish the odorants that attract or inhibit oviposition, so as to locate a suitable oviposition substrate [33,34]. Our study found that the egg-laying numbers of FHo were significantly lower than those of the same mating state FWT, regardless of whether they mate with MWT or MHo, suggesting that the egg-laying decisions of females were altered under conditions where the olfactory system is almost lost (Figure 5A). This phenomenon has been consistently reported in other insects. For example, in *Nilaparvata lugens*, homozygous mutant females lacking *NlugOrco* experienced a 23.6% reduction in egg-laying compared to WT females [35]. In the case of *S. frugiperda*, WT females typically laid approximately 1746.2 eggs, whereas *SfurOrco^−/−^* homozygous mutant females laid only 1332.58 eggs [18]. Similarly, the knockout of the *Orco* gene can significantly reduce the egg-laying numbers of two ants, namely *Ooceraea biroi* and *Harpegnathos saltator* [17,36].

Combined with previous studies, the decrease in the egg production of mutant females may be related to two reasons: (1) It may be that the odorant receptor is involved in the regulation of the occurrence of female eggs, resulting in a decrease in the number of eggs. (2) It is possible that the egg-developments of *Orco^−/−^* mutant females are not affected, but the olfactory system involved in the perception of oviposition stimulation is defective, resulting in the egg-laying signals cannot be received. However, it is not clear how olfaction affects the oviposition of *E. corollae*, and further study is needed to elucidate the molecular mechanism of detection to oviposition stimuli. In addition, this study showed that the hatching rate of eggs produced by females mated with MHo was extremely reduced. It may be related to the motility of male sperm, as a study found that the *Orco* gene of *Anopheles gambiae*, *Nasonia vitripennis,* and *Aedes albopictus* was located on the flagella of spermatozoa and mediated the activation of spermatozoa [37]. Hence, another possible reason for the decrease in the reproductive ability of *EcorOrco^−/−^* homozygous mutants of *E. corollae* is that the deletion of *Orco* gene leads to the decrease in the sperm motility of male hoverfly.

Taken together, our results provide in vivo evidence that the *Orco* gene of the *E. corollae* is involved in chemoreception to odorants in their surroundings, and affects their mating, egg-laying behaviors, and their fertilities. The olfactory recognition mechanism of hoverflies mediated by *Orco* gene was preliminarily elucidated. 

## 4. Materials and Methods

### 4.1. Insects

Adults *E. corollae* were collected from the Langfang Experiment Station of the Institute of Plant Protection, Chinese Academy of Agricultural Sciences (CAAS), Langfang, Hebei province, China (116.60° E, 39.50° N). *E. corollae* was reared in our laboratory with *Acyrthosiphon pisum* (*A. pisum*) and *Megoura crassicauda* (*M. crassicauda*) at the larval stage and pollen and 10% honey solution at the adult stage. The rearing conditions were maintained at a temperature of 25 ± 1 °C, a relative humidity of 60 ± 5%, and a photoperiod of 12:12 (light:dark).

*A. pisum* and *M. crassicauda* were reared on broad bean plants (*Vicia fabae* L.) at the Institute of Plant Protection, Chinese Academy of Agricultural Sciences, Beijing, China. These aphid species were reared under controlled conditions with a temperature of 20 ± 2 °C, a relative humidity of 70 ± 5%, and a photoperiod of 16:8 (light:dark).

### 4.2. Single Guide RNA (sgRNA) Preparation of EcorOrco

The exons and introns of the *EcorOrco* gene were annotated utilizing transcriptome and genome data previously reported for *E. corollae* [38,39]. Furthermore, to verify the genome sequence of *EcorOrco* gene, genomic DNA (gDNA) was individually extracted from 8 adult specimens using the TIANamp Genomic DNA Kit (TIANGEN, Cat#DP304-03, Beijing, China), following the manufacturer’s instructions. To identify suitable gene knockout targets, a pair of gene-specific primers were designed to amplify an 890 bp-long DNA fragment using the gDNA as a template. The forward primer was located in exon 1, and the sequence was 5′-GCCTTGTAGCCGACCTGATG-3′; the reverse primer was situated in exon 3 with the sequence: 5′-CAACTCCTGCCAAAGACGAATAAA-3′. The target DNA fragment was amplified in a 25 μL PCR reaction mixture composed of 12.5 μL of 2 × Phanta Flash Master Mix (Dye Plus), 1 μL of each primer, 1 μL of gDNA template, and 9.5 μL of ddH_2_O. The PCR was performed at 98 °C for 30 s, 35 cycles of 98 °C for 10 s, 59 °C for 5 s, and 72 °C for 5 s, followed by a final extension of 72 °C for 1 min. The PCR products were subsequently sent for sequencing, utilizing both the forward and reverse primers.

The selection of target gene editing sites was based on the principle of 5′-(19N) NGG-3′, and we selected two conserved sequences located in exon 2 and exon 3 as the target sites for gene editing. To ensure the specificity of these target sites and evaluate the potential for off-target effects, we conducted BLAST searches against the *E. corollae* genome. The sgRNAs were synthesized according to the manufacturer’s instructions (GeneArt Precision gRNA Synthesis Kit, ThermoFisher Scientific, Pittsburgh, PA, USA). Before synthesis, we assembled a forward primer (5′-GAAATTAATACGACTCACTATAG + target sequence-3′) and a reverse primer (5′-TTCTAGCTCTAAAAC + target sequence reverse complement-3′) to prepare DNA templates for gRNA synthesis with the Tracr Fragment + T7 Primer Mix by PCR under the following conditions: 98 °C for 10 s, followed by 32 cycles of 98 °C for 5 s and 55 °C for 15 s, concluding with 72 °C for 1 min. Subsequently, the sgRNAs were generated through in vitro transcription (IVT) reactions. The DNA templates were then eliminated using DNase I, and the synthesized products were purified by the gRNA Clean Up Kit (GeneArt Precision gRNA Synthesis Kit, ThermoFisher Scientific, Pittsburgh, PA, USA). Each sgRNA was then diluted to a concentration of 300 ng/μL in nuclease-free water and stored at −70 °C.

### 4.3. Early Embryos Microinjection

Freshly laid eggs were collected from over fifty mated *E. corollae* females and promptly affixed to a microscope slide using double-sided adhesive tape. These procedures were completed within thirty minutes after oviposition.

For microinjection, a Cas9/sgRNA complex was prepared, comprising 300 ng/μL of sgRNA1, 300 ng/μL of sgRNA2, and 300 ng/μL of Cas9 protein. This complex was injected into the prepared eggs using the FemtoJet and InjectMan NI 2 microinjection system (Eppendorf, Hamburg, Germany). In total, approximately 1300 eggs were injected and incubated at 25 °C and 60% RH for 2–3 days until hatching.

### 4.4. EcorOrco Mutant Identification

A total of 53 larvae successfully hatched from the injected eggs. Subsequently, the G0 adult *E. corollae* individuals were subjected to crossbreeding with one another to establish the G1 generation. Each pair of G0 adults was placed in separate plastic cups, which contained a diet consisting of pollen and a cotton ball soaked in a 10% honey solution. Additionally, a broad bean seedling infested with aphids was provided, allowing the *E. corollae* individuals to mate and lay eggs.

Following oviposition, the whole bodies of the G0 adults were utilized for genomic DNA extraction, following the protocol described in Section 4.3. We successfully amplified an 890 bp DNA fragment, encompassing the two sgRNA sites. The PCR products were confirmed by running them on a 1% agarose gel, and the products with a long DNA fragment deletion were sent to sequencing with the forward primer. Based on the obtained sequences, only the larvae generated by parents containing the mutation were reared. These larvae were further crossed and reared in individual pairs to produce the G2 progeny. The mutation was detected using the same procedure, and ultimately, the homozygous mutation line was established in the G5 progeny. Subsequently, we conducted an amino acid sequence alignment, comparing the wild-type (WT) and the homozygous mutation lines of EcorOrco using DNAMAN 8 software (Lynnon Biosoft, Quebec, QC, Canada). Furthermore, to predict the transmembrane domains within the EcorOrco protein, we utilized the TMHMM-2.0 website, accessible at the following link (https://services.healthtech.dtu.dk/service.php?TMHMM-2.0, accessed on 9 March 2022).

### 4.5. Immunohistochemisty

To prepare the antisense probe, first generate an 802 bp double-stranded fragment of *Orco* from the plasmid containing the full-length cDNA of *EcorOrco*, then ligate the 802 bp fragment of *EcorOrco* into a pspt18 vector that contains promoters for T7 (upstream) and SP6 (downstream) RNA polymerases adjacent to the inserted DNA using T4 DNA ligase. Utilize the T7/SP6 RNA transcription system to synthesize the antisense probe. T7 RNA polymerase is used to transcribe double-stranded RNA for biotin-labeled antisense probe for this *Orco* gene using linearized recombinant plasmids as templates in vitro. The probe is stored at −80 °C.

Antennae were removed and embedded in Tissue-Tek OCT (Sakura Finetek, Torrance, CA, USA), then promptly frozen them to fix these tissues. Antennae were sectioned at 10 μm at −24 °C, and these slices were then individually mounted on slides one by one. Slides were fixed in 4% paraformaldehyde, then followed by sequentially washing the slides in 1 × phosphate-buffered saline (PBS), 0.2 M HCl and 1 × PBS with 1% Triton X-100, respectively [40]. Rinse the slides twice in 1 × PBS and finish with a rinse in a formamide solution. To prepare the antisense probe for detection, dilute biotin-labeled EcorOrco probes in the hybridization buffer. Add 1 µL of probe to 99 µL of hybridization buffer per slide. Perform hybridization by draining the slides and adding 100 µL of diluted antisense probe to the tissue sections. Place nuclease-free coverslips (24 mm × 50 mm) on top of the tissue sections. Arrange the covered slides horizontally in a humid box (300 × 180 × 50 mm^3^) and incubate for 20 h.

After hybridization, carefully remove the coverslips. Wash the slides twice in 0.1 × saline sodium citrate (SSC). Add 1 mL of 1% blocking reagent in TBS supplemented with 0.03% Triton X-100 on each slide, and incubate [41]. For chromogenic detection, dilute 750 U/mL of anti-biotin streptavidin HRP-conjugated antibody in blocking reagent in TBS and add 100 µL of the HRP solution to each covered slide. Incubate the slides in a humid box. For staining, add 100 µL of Tyramide diluent to each covered slide and incubate. To slide, drop a few drops of slow antifade reagent with DAPI, then cover the coverslips with tweezers to fix nail polish around them.

Observe the tissue sections using a confocal microscope (LSM 980, Carl Zeiss Microscopy GmbH, Jena, Germany). Biotin-labeled genes should be observed under 517 nm light, displaying a green color, while DAPI-labeled genes should be visible under 465 nm light, presenting a blue color. Utilize the 10 × objective lenses to observe the results of detection.

### 4.6. Chemicals and EAG Recording

The 15 odorant compounds utilized in this study were all purchased from Sigma-Aldrich. Detailed information regarding the names, types, and CAS numbers of these tested odorants is provided in Table 1. Before EAG assays, the chemicals were dissolved in paraffin oil to a concentration of 10 μg/μL and stored at −20 °C until use.

The antennae of mutant and WT hoverflies (3–4 days after eclosion) were removed at their base. Subsequently, these excised antennae were positioned between two glass electrodes filled with a 0.1 mol/L KCl. In each test, a 10 μL test solution (or solvent) was applied onto a rectangular piece of filter paper (0.4 cm × 4 cm), which was subsequently inserted into a Pasteur pipette. EAG recordings were conducted following the method described in Wang et al [20]. The experimental setup was maintained in a controlled environment with continuous humidified airflow, regulated by a stimulus controller (CS-55, Syntech, Kirchzarten, Germany), at a flow rate of 30 mL/s. Odor stimulations were performed through 0.2 s pulses at a 10 mL/s airflow rate, with 30 s intervals between stimulations. EAG signals were amplified using a 103AC/DC headstage preamplifier (Syntech) and subsequently recorded with an Intelligent Data Acquisition Controller (IDAC-4-USB, Syntech). The recorded signals were closely monitored and subjected to analysis using Syntech EAG-software (version 2.0) [20]. To ensure precision and reliability, each antenna underwent two stimulations with paraffin oil and the average values obtained by paraffin oil were subtracted from the data.

### 4.7. Behavioral Experiments

#### 4.7.1. Y-Tube Olfactometer Assay for Adults

The behavioral response of adult hoverflies to EBF was assessed using a glass Y-tube olfactometer, following the procedure described in Li et al [19]. This olfactometer featured a consistent diameter of 2 cm, with both the trunk and branch measuring 14 cm in length. The branch angle was set at 90°, and the experiments were conducted at a controlled environment of 23 ± 1 °C and 40–60% relative humidity.

A continuous flow of humidified air filtered through activated granular carbon was supplied at a flow rate of 0.5 L/min. A piece of filter paper was positioned at the terminus of each branch within the Y-tube olfactometer. One of these filter paper pieces was impregnated with EBF at a concentration of 20 μg/μL, dissolved in hexane, while the other filter paper served as a control and was treated with pure hexane. Before commencing the experiments, the tests of 2- to 3-day-old naïve male and female hoverflies were prepared following the methods described in Wang et al [20]. Following this preparation, individual hoverflies were introduced into the distal end of the olfactometer’s trunk and observed during a period of 10 min. In the course of observation, a hoverfly was considered to have made a choice if it was within one-third of the branch length and remained in that position for at least 30 s. Otherwise, it was recorded as not having made a choice. To minimize any positional bias, the positions of the Y-shaped olfactometer’s branches were interchanged after each three replications. After testing each set of six hoverflies, the Y-shaped olfactometer was cleaned using ethanol and subjected to heating at 180 °C for 3 h before reuse. All bioassays were conducted during daylight hours between 8:00 and 17:00.

#### 4.7.2. Two-Way Choice Assay for Larvae

The choice test for larvae was conducted in plastic Petri dishes with a diameter of 14.5 cm, following the procedure outlined in Wang et al [20]. Two circular holes were drilled in the Petri dishes, each having a diameter of 1.5 cm. These holes were positioned at opposite ends along the diameter of the dishes. Beneath these holes, we positioned two little Petri dishes measuring 5.5 cm in diameter. Rubber septa, affixed with plasticine onto the small Petri dishes, were loaded with 100 μL of the respective test solutions. The treatment was EBF at a concentration of 0.2 μg/μL, dissolved in hexane, and hexane serving as the control. Ten-second-instar larvae were gently placed in the center of the larger Petri dish. The dish area was divided into two parts, delimited by the arcs of circles centered around the odor spots (with a radius of 5.5 cm). The number of larvae in each area was recorded at 30 min. This process was repeated six times for robust data collection.

#### 4.7.3. Mating and Fecundity Tests

To investigate the role of the *Orco* gene in *E. corollae* mating and oviposition processes, we conducted behavioral experiments in carefully controlled environmental conditions. These experiments were conducted at a temperature of 26 ± 1 °C and a relative humidity of 50 ± 10%. We selected 5-day-old virgin adult *E. corollae* as subjects for the mating test.

The mating experiments were organized into four distinct groups: 1. female WT (FWT) × male WT (MWT); 2. female WT (FWT) × male homozygote (MHo); 3. female homozygote (FHo) × male WT (MWT); 4. female homozygote (FHo) × male homozygote (MHo). Pairs of male and female *E. corollae * with the appropriate genetic profiles were housed in plastic containers measuring 10.5 cm in height and 8 cm in diameter. We employed a SONY camera to record their mating behavior during the period between 8:00 and 18:00. Subsequently, we analyzed their mating behavior using video software and calculated the mating rate.

To evaluate the influence of the *Orco* gene on egg laying in female hoverflies, we randomly selected 12 mated female *E. corollae* from groups 1–4, representing different genotypes. Each selected female was then placed into an individual plastic container and provided with a diet of canola pollen and a 10% honey solution. We provided each female with a plant infected by aphids for oviposition and recorded the egg-laying number by each individual over consecutive days.

### 4.8. Statistical Analysis

The relative EAG value of the antenna was calculated. The specific formula is as follows: the relative EAG response value = |response value by tested odorant − mean value by paraffin oil|. The preference index (PI) of the larvae to EBF was calculated using the formula: PI = (P − C)/(P + C), where P signifies the number of larvae that entered the EBF zone, and C signifies the number of larvae that entered the solvent zone. The relative EAG response values between different genotypes to the odorant, the preference index of larvae for EBF, and the number of egg were analyzed using Student’s *t*-test (α = 0.05). The preference of different genotypes adults of the same sex to EBF, the mating rate, and the hatching rate were analyzed using chi-square test for 2 × 2 contingency tables.

All statistical analyses were conducted using SPSS 25.0 software (SPSS Inc., Chicago, IL, USA) and presented as means ± SEM. Figures were generated using GraphPad Prism 7.0 software (San Diego, CA, USA).

## Figures and Tables

**Figure 1 ijms-24-17284-f001:**
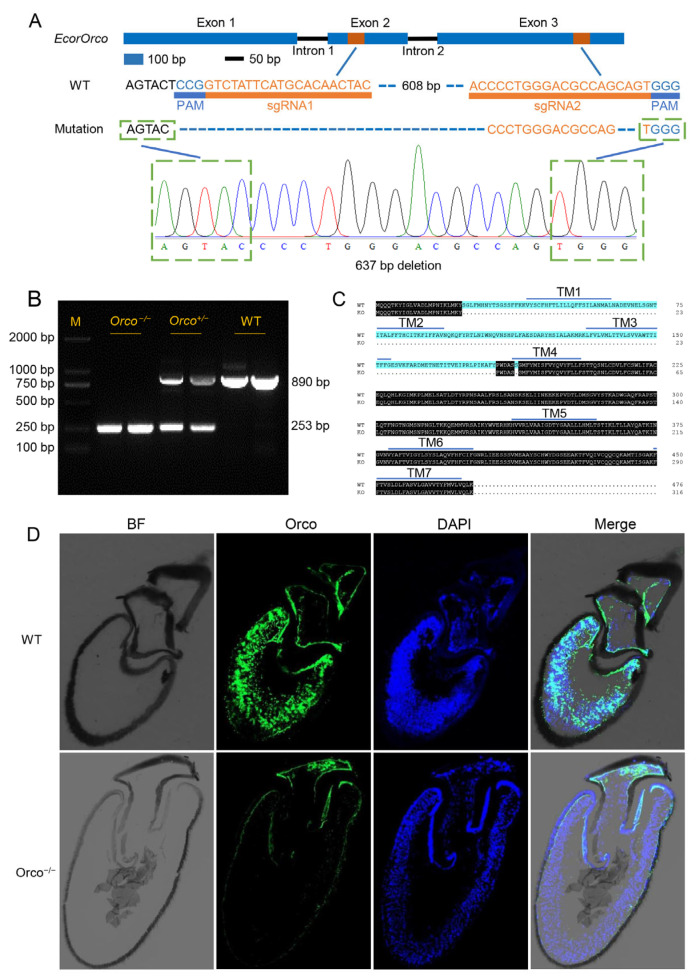
*EcorOrco* gene was knocked out by CRISPR/Cas9 technique. (**A**) Schematic diagram of the sgRNA-targeted sites of *EcorOrco*. Two single-guide RNA (sgRNA) targets (in orange) were chosen and designed, with their respective PAM sequences highlighted in blue. Genome editing took place at both of these sites, leading to the deletion of a 637 bp DNA fragment in the genome. (**B**) The genotypes were identified using PCR and agarose gel electrophoresis. (**C**) The alignment of the EcorOrco sequence in both the wild-type (WT) and Orco knock-out (KO) lines reveals the presence of seven transmembrane domains, indicated by blue lines. (**D**) Top: antennal morphology of WT in the bright field (gray), antennal section of WT showing *Orco* immunostain (green), DAPI counterstain (blue), and the merged image. WT possesses a dense region of Orco-positive ORNs in the center of the antenna. Bottom: antennal section of *EcorOrco^−/−^* mutant. *EcorOrco^−/−^* mutant also possesses a dense region of Orco-positive ORNs in the center of the antenna. Scale bars, 20 mm.

**Figure 2 ijms-24-17284-f002:**
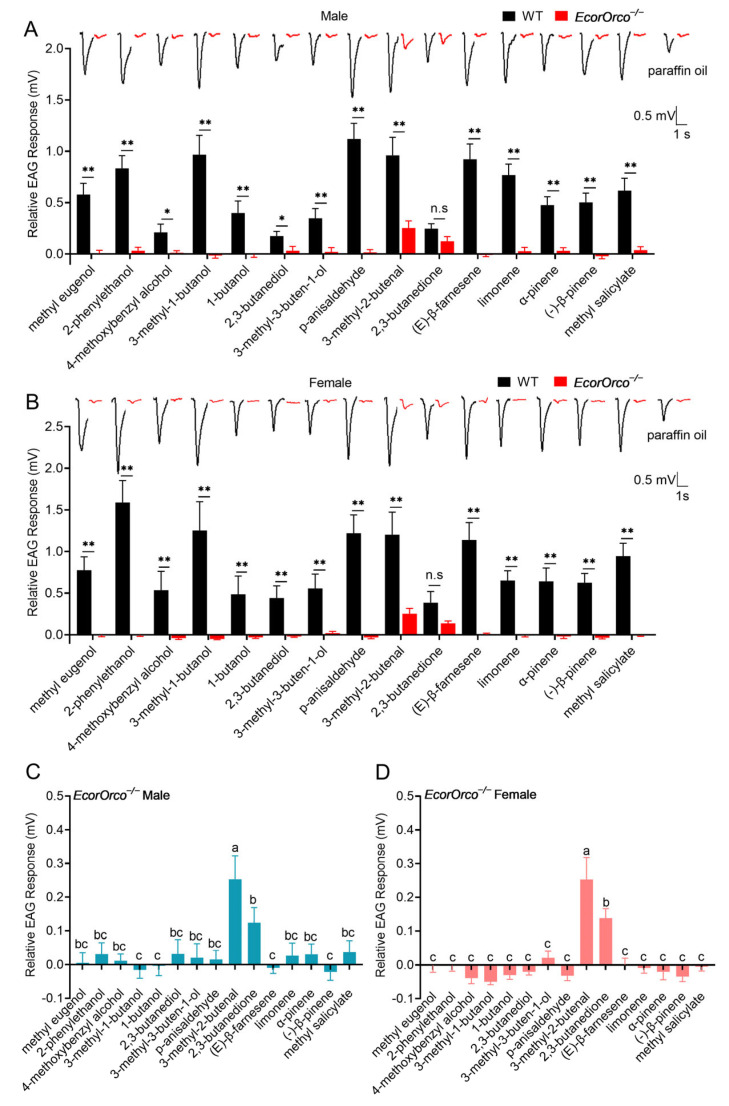
Electroantennogram responses of WT and *EcorOrco^−/−^* adults in *E. corollae* to 15 odorants. (**A**,**B**) Traces of EAG responses and the relative EAG responses tested in male (**A**) and female (**B**) *E. corollae*, (**C**,**D**) The relative EAG responses tested in *EcorOrco^−/−^* male (**C**) and *EcorOrco^−/−^* female (**D**) *E. corollae.* No significant differences were observed in the response to 2,3-butanedione between the WT and *EcorOrco^−/−^* hoverflies, while EAG signals were fully impaired in *EcorOrco^−/−^* hoverflies in response to other tested odorants when compared with WT hoverflies. EAG responses are means ± SEM ((**A**,**B**): Student’s *t*-test, n.s, *p* > 0.05, * *p* < 0.05, ** *p* < 0.01; (**C**,**D**): Duncan’s multiple range test, and different letters indicate significant differences at α = 0.05 level, *n* = 15).

**Figure 3 ijms-24-17284-f003:**
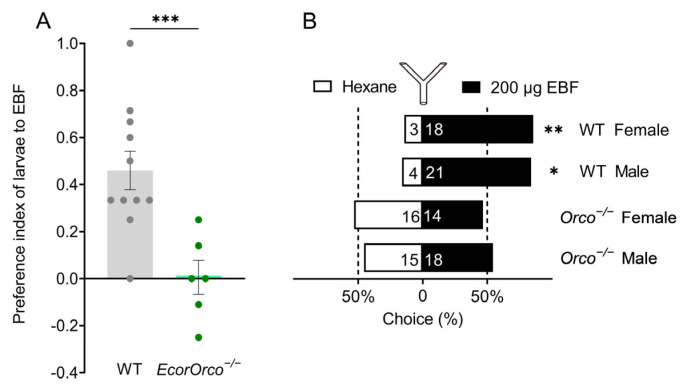
*EcorOrco^−/−^* from both larvae and adults lost its behavioral preference to EBF. (**A**) The behavioral preference of the different genotypes of *E. corollae* larvae to EBF. Compared with the WT, the *EcorOrco^−/−^* hoverflies lost the behavioral preference to 20 μg EBF (Student’s *t*-test, *** *p* < 0.001, *n* = 6–10). Data are plotted as mean ± SEM. (**B**) Behavioral choice of WT and *EcorOrco^−/−^* hoverflies to 200 μg EBF in a Y-tube olfactometer. Both sexes of WT hoverflies were significantly attracted by EBF (chi-square test, * *p* < 0.05, ** *p* < 0.01), but lost preference in *EcorOrco^−/−^* hoverflies (chi-square test, *p* > 0.05).

**Figure 4 ijms-24-17284-f004:**
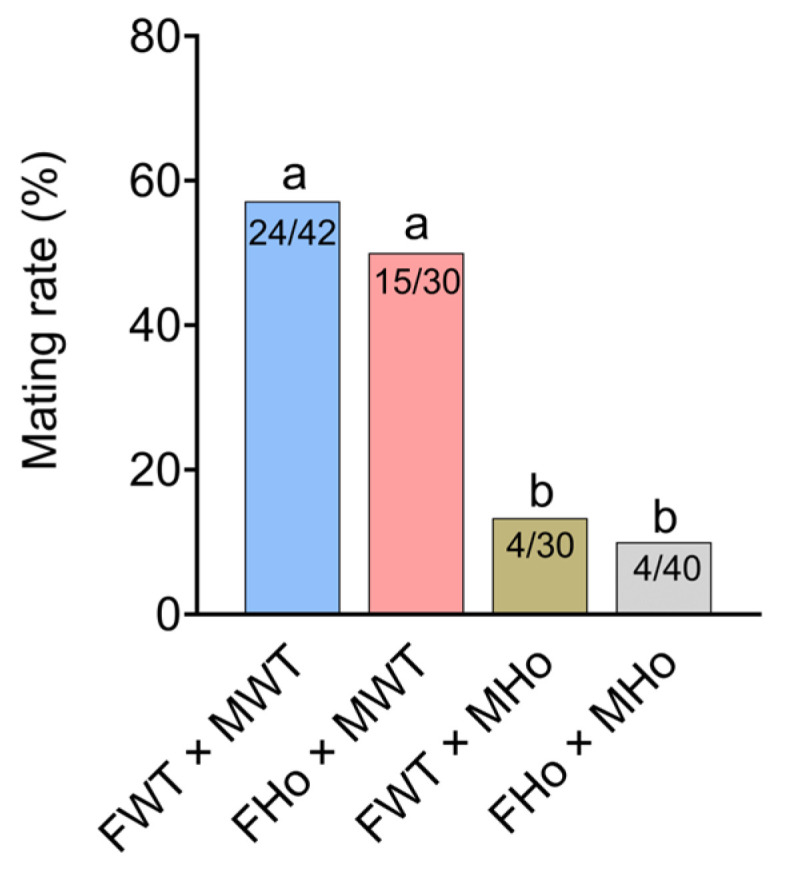
The mating behavior of WT and *EcorOrco^−/−^* mutant lines. The mating rate of different genotypes were examined in four combinations, including female WT (FWT) × male WT (MWT), female homozygote (FHo) × male WT (MWT), female WT (FWT) × male homozygote (MHo), and female homozygote (FHo) × male homozygote (MHo). The total number of pairs and the number of successfully mated pairs are marked in each column. Significant differences between four combinations are calculated by chi-square test. Different letters indicate significant differences at the α = 0.05 level.

**Figure 5 ijms-24-17284-f005:**
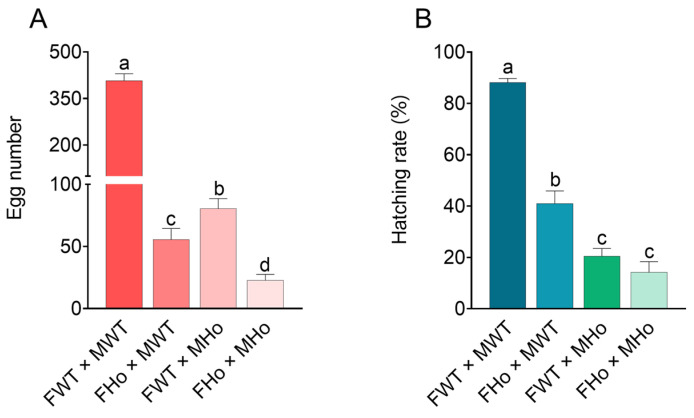
The fecundity and hatching rate of WT and *EcorOrco^−/−^* mutant lines. (**A**) The cumulative egg-laying number of the successfully mated females in the four combinations. (**B**) Hatching rate of eggs laid by the successfully mated females in the four combinations. Significant differences between four combinations are calculated by Duncan’s multiple range test. Different letters indicate significant differences at the α = 0.05 level. Error bars indicate SEM (*n* = 12).

**Table 1 ijms-24-17284-t001:** The list of chemicals used in this study.

Chemicals	Types	CAS Numbers
methyl eugenol	Aromatics	93-15-2
2-phenylethanol	Aromatics	1960-12-8
4-methoxybenzyl	Aromatics	105-13-5
p-anisaldehyde	Aromatics	123-11-5
3-methl-2-butenal	Aromatics	107-86-8
methyl salicylate	Aromatics	119-36-8
3-methyl-1-butanol	Aliphatics	123-51-3
1-butanol	Aliphatics	71-36-3
2,3-butanediol	Aliphatics	513-85-9
3-methyl-3-buten-1-ol	Aliphatics	763-32-6
2,3-butanedione	Aliphatics	431-03-8
(*E*)-β-farnesene (EBF)	Terpenes	18794-84-8
limonene	Terpenes	138-86-3
*α*-pinene	Terpenes	80-56-8
(-)-β-pinene	Terpenes	18172-67-3

## Data Availability

No new data were created or analyzed in this study. Data sharing is not applicable to this article.

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
