# Peer review of "Mutagenesis of Odorant Receptor Coreceptor Orco Reveals the Odorant-Detected Behavior of the Predator Eupeodes corollae"

_ijms, 2023, doi:10.3390/ijms242417284_

Round 1

Reviewer 1 Report

Comments and Suggestions for Authors

Reviewer comment’s #:

The manuscript title addresses “Mutagenesis of odorant receptor coreceptor Orco reveals the odor- ant-detected behavior of the predator Eupeodes corollae” The title of the manuscript is broader than the subject incorporated in it. Here, the authors try to understand role of EcorOrco gene in hoverflies. The manuscript is well-written, which I overall accept. The parameters chosen for strategy building are traditional and informative. However, I do suggest certain things that need clarification to support and strengthen the conclusion.

Abstract: Needs improvement with essential points to be highlighted.

Introduction: Please elaborate and update with the latest information which is missing.

Please provide some details about insect hormones that play a key role in reproduction and try to link them with receptors, which can possibly provide more clarity.

Please update additional details in section 2.2 and 4.6.

What did the Electroantennogram (EAG) results show regarding the electrophysiological response of the adult mutant to odorants? Please provide some pictures for clarity.

How did both larvae and adults of hoverflies behave in the two-way choice assay and the glass Y-tube olfactometer after the knockout of the EcorOrco gene? Explain it clearly.

The conclusion part is very weak and needs to be reorganized because a few parts of explanations are not clear and are missing the latest references.

General comments: There are few typos throughout the manuscript. In fact, the manuscript needs to be clearer on the chosen subject, and perhaps some additional approaches are needed. The content presented here is more theatrical than expected. There are many places that need to be updated with the latest references. Please make the manuscript more unique by adding more recent details. I suggest the authors go for English corrections with a native speaker or a professional company.

The manuscript is interesting, but strategic connectivity to the proposed hypothesis is lacking at certain places. Since I found some difficulty in reading and understanding certain parts of the manuscript, the article needs some corrections and certain details which need to be incorporated. I do think that the manuscript contains important issues, interesting approaches, and techniques, which can lead to an understanding of the role of the EcorOrco gene in hoverflies and can be used for pest control strategies. Therefore, I consider this manuscript suitable for publication after major revision in IJMS.

Comments on the Quality of English Language

English corrections needed.

Author Response

The manuscript title addresses “Mutagenesis of odorant receptor coreceptor Orco reveals the odor- ant-detected behavior of the predator Eupeodes corollae” The title of the manuscript is broader than the subject incorporated in it. Here, the authors try to understand the role of EcorOrco gene in hoverflies. The manuscript is well-written, which I overall accept. The parameters chosen for strategy building are traditional and informative. However, I do suggest certain things that need clarification to support and strengthen the conclusion

Abstract: Needs improvement with essential points to be highlighted.

Answer: We believe that this abstract is a detailed, concise and clear description of the research results and significance of the current version. Of course, there will be some problems that we overlook and need to be improved, please specify what needs to be revised. We have revised some of the statements based on our understanding.

Introduction: Please elaborate and update with the latest information which is missing.

Answer: We have updated some references, but we don't know how to modify the general question you raised. Please be more specific, such as what is the deficiency of the sentence mentioned in which paragraph and which line, so that we can modify it.

Please provide some details about insect hormones that play a key role in reproduction and try to link them with receptors, which can possibly provide more clarity.

Answer: In fact, it's not really clear here what kind of correlation you're referring to and how relevant is the content of this study. Our study provides some in vivo evidence for the role of Orco in the olfactory recognition process of hoverflies. However, since no studies on sex pheromones in hoverflies have been reported so far, it is difficult to provide the correlation of Orco participation in pheromone recognition. Future studies are expected to resolve the correlation you mentioned, but this correlation is obvious.

Please update additional details in section 2.2 and 4.6.

Answer: We have modified and refined it. Please see Line 131-133, 428-435.

What did the Electroantennogram (EAG) results show regarding the electrophysiological response of the adult mutant to odorants? Please provide some pictures for clarity.

Answer: The EcorOrco−/− mutants displayed a nearly complete loss of EAG responses to the tested chemicals, except for 2,3-butanedione and 3-methyl-2-butenal. Because they may have other olfactory pathways, such as IR, involved in detection. this result is consistent with what has been reported in Drosophila melanogaster. We have added the EAG traces of WT and EcorOrco-/- mutant to all tested chamicals.

How did both larvae and adults of hoverflies behave in the two-way choice assay and the glass Y-tube olfactometer after the knockout of the EcorOrco gene? Explain it clearly.

Answer: We have described in line 80 of the introduction that (E) -β-farnesene (EBF) is attractive to E. corcollae, and EBF is an aphid alarm pheromone component that helps E. corcollae accurately locate prey aphids. Therefore, we used EBF compound as a positive control to detect the olfactory response of EcorOrco-/- mutants to EBF. When EcorOrco gene was knocked out, both larvae and adults lost their behavioral preference for EBF. These results indicate that Orco gene plays an important role in the process of olfactory recognition.

The conclusion part is very weak and needs to be reorganized because a few parts of explanations are not clear and are missing the latest references.

Answer: The last paragraph in the discussion is the conclusion part. This is a highly concise summary of the results of the full text, and should not include the references. I consider what you're referring to here is the discussion section. If there is any inappropriateness, please point it out clearly and define them so that we can make further corrections.

General comments: There are few typos throughout the manuscript. In fact, the manuscript needs to be clearer on the chosen subject, and perhaps some additional approaches are needed. The content presented here is more theatrical than expected. There are many places that need to be updated with the latest references. Please make the manuscript more unique by adding more recent details. I suggest the authors go for English corrections with a native speaker or a professional company.

Answer: We have updated the references. Besides, this manuscript has been revised and polished by the English native speaker.

The manuscript is interesting, but strategic connectivity to the proposed hypothesis is lacking at certain places. Since I found some difficulty in reading and understanding certain parts of the manuscript, the article needs some corrections and certain details which need to be incorporated. I do think that the manuscript contains important issues, interesting approaches, and techniques, which can lead to an understanding of the role of the EcorOrco gene in hoverflies and can be used for pest control strategies. Therefore, I consider this manuscript suitable for publication after major revision in IJMS.

Answer: Thanks for your suggestions.

Reviewer 2 Report

Comments and Suggestions for Authors

Authors Wu et al. presented a study of revealing the functions of Orco by CRISPR/cas 9 in hoverfly, a natural enemy of aphids. Authors first knocked out the Orco from the genome of hoverfly using CRIPSR/Cas 9 technology. After confirming, authors performed electroantennogram and behavioral assays and found that both larva and adult hoverflies without Orco failed to response to 15 different odorants and lost preference to aphid alarm pheromone EBF. The mating behavior of Orco mutants showed that Orco gene plays an important role in mate mating process. In females, although Orco knockout did not affect female mating, lack of Orco reduce eggs-laying amount and the hatching rate, suggesting Orco may play roles in the development of embryos. The manuscript is very well designed, and the data presentation and interpretation are clear and concise. I only have some minor comments as below:

1.     Line 144-145, why “the EcorOrco mutants did not significantly reduce the EAG response to 2,3,-butanedione, suggesting the absence of relevant of off-target effects”? It does not make sense.

2.     Line 209, delete “eggs and” because of redundance.

3.     Did author perform any off-target effect analysis or try any strategies to minimize off-target effects?

Author Response

Line 144-145, why “the EcorOrco mutants did not significantly reduce the EAG response to 2,3,-butanedione, suggesting the absence of relevant of off-target effects”? It does not make sense.

Answer: Thanks for your valuable suggestions. We have revised it. Please see Line 147.

Line 209, delete “eggs and” because of redundance.

Answer: We have revised it. Please see Line 215.

Did author perform any off-target effect analysis or try any strategies to minimize off-target effects?

Answer: Thanks for your suggestions. In order to prevent the off-target effect, we conducted BLAST analysis against the E. corollae genome, and found no potential off-target sites.

Round 2

Reviewer 1 Report

Comments and Suggestions for Authors

Reviewer #:

Now, this is a very well-conceived and written paper.

The author incorporated the particulars in the present revised version of the manuscript.

Please revise the incorporation perfectly. I found typos in the incorporated text please correct them.

Finally, please check the references carefully if all are according to the format of the IJMS.

With such changes, I agree this article be published in IJMS.